# Optimizing Protein Fiber Spinning to Develop Plant-Based Meat Analogs via Rheological and Physicochemical Analyses

**DOI:** 10.3390/foods12173161

**Published:** 2023-08-23

**Authors:** Kartik Joshi, Elnaz Shabani, S. M. Fijul Kabir, Hualu Zhou, David Julian McClements, Jay Hoon Park

**Affiliations:** 1Department of Plastics Engineering, University of Massachusetts Lowell, Lowell, MA 01854, USA; kartik_joshi@student.uml.edu (K.J.); eshaban@alumni.ncsu.edu (E.S.); smfijul_kabir@uml.edu (S.M.F.K.); 2Department of Food Science, University of Massachusetts Amherst, Amherst, MA 01003, USA; hualuzhou@umass.edu (H.Z.); mcclemen@umass.edu (D.J.M.)

**Keywords:** rheology, soy protein, meat muscle, extrusion, fiber

## Abstract

The substitution of meat products in the human diet with plant-based analogs is growing due to environmental, ethical, and health reasons. In this study, the potential of fiber-spinning technology was explored to spin protein fiber mimicking the structural element of meat muscle for the purpose of developing plant-based meat analogs. Overall, this approach involved extruding fine fibers and then assembling them into hierarchical fibrous structures resembling those found in whole muscle meat products. Considering the nutritional facts and to help build muscle fiber, soy protein, polysaccharide (pectin, xanthan gum, or carrageenan), plasticizer (glycerol), and water were used in the formulations to spin into fibers using an extruder with circular orifice dies. Extrudability and thermal and rheological properties were assessed to characterize the properties of the spun fiber. The extrusion trials showed that the presence of the polysaccharides increased the cohesiveness of the fibers. The properties of the fibers produced also depended on the temperature used during extrusion, varying from pasty gels to elastic strands. The extrudability of the fibers was related to the rheological properties (tan δ) of the formulations. This study demonstrated that fiber-spinning technology can be used to produce fibrous materials from plant-derived ingredients. However, the formulation and operating conditions must be optimized to obtain desirable physicochemical and functional attributes in the fibers produced.

## 1. Introduction

The production of animal-sourced foods, such as meat, eggs, and milk, has been reported to cause appreciable environmental damage, such as increased greenhouse gas emissions, pollution, biodiversity loss, land use, and water use [1,2,3]. Therefore, there is considerable interest in reducing the total amount of these foods, especially meat, in the human diet to improve its sustainability and reduce its adverse environmental impacts [4,5]. As a result, the food industry is developing a range of plant-based meat analogs to replace real meat products. The consumer acceptability of these plant-based products depends on how well they resemble the physiochemical, functional, organoleptic, and nutritional attributes of real meat products [6,7,8]. Also, the convenience, cost, and availability of these products should be similar to or better than those of real meat products [9].

There are several approaches to develop meat alternatives besides plant-based meats such as precision fermentation or lab-grown meat. Among these, plant-based meat analogs are often considered to have advantages over other technologies because of the cost-effectiveness and scalability [5,10]. In addition to reducing the negative environmental impacts of the livestock industry, plant-based meat analogs may also have other advantages, including improving human health and animal welfare and preserving cultural and religion diversity [8,9,11,12]. The methods of growing meat alternatives usually involve applying high temperatures, pressures, and shear forces to achieve an optimized mixture of plant-derived proteins, polysaccharides, and other ingredients to create fibrous structures that mimic those found in animal meat [13,14,15].

The main hurdle towards the more widespread adoption of plant-based meat analogs is being able to accurately match the desirable appearance, texture, and flavor of real meat products [16]. There have already been several successful plant-based meat analogs introduced onto the market, but these are mainly designed to mimic the properties of comminuted meat products, like sausages, burgers, and nuggets [17,18]. However, creating products that simulate the properties of whole muscle meat (such as beef steaks, chicken breasts, or pork chops) still remains as a challenge, due to the complex hierarchical structure of the protein fibers within these products [19,20].

Consequently, there is a pressing need and tremendous interest to identify economic and scalable processing methods of creating fibrous structures from plant-based ingredients with a view to producing whole muscle meat analogs [21]. Approaches include shear cell, 3-D printing technologies, and fiber spinning/extrusion [22]. While shear cell and 3-D printing technologies have been explored to some extent for producing meat alternative, fiber spinning/extrusion techniques still need more attention [23]. Although, some researchers have already attempted to produce fibers from soy proteins using wet spinning, batch dry spinning, and continuous extrusion processes [24,25,26,27]. However, most of these studies were not intended for application in meat analogs, except a few recent studies [15,22,28,29]. The produced fibers lack the required properties to mimic whole cut meat, and hence, the formulation and process parameters need to be optimized to resemble meat-like properties with due consideration of scalability and economic viability [30].

Herein, the present research aims to develop individual soy-protein-based fibers with improved structural and physicochemical properties using fiber extrusion technology. To match the plant-based meat’s nutritional profile to real meat (for example, 15 to 30 wt% of protein and 50–85 wt% moisture in real meat [31]), as well as to simplify the fiber extrusion process, the ingredients in the formulation were judiciously selected. Based on abundance, wide application, and having good nutritional value, soy protein was used as the main functional ingredient for protein supply [32]. Other additives include food-grade polysaccharides (pectin, xanthan gum, and carrageenan) to obtain high-molecular-weight polymers with extended structures for improving the extrudability of plant proteins [33], plasticizers such as buffer solution (water) for maintaining moisture and pH, and glycerol for moisture retention.

## 2. Materials and Methods

### 2.1. Materials

Soy protein isolate (SPI) (Profam 974) in powder form was supplied by the ADM Company (Chicago, IL, USA). Phosphate buffer and glycerol were obtained from the Sigma-Aldrich Company (St. Louis, MO, USA). High-methoxy pectin, xanthan gum, and carrageenan in granular powder form were obtained from TIC Gums (Belcamp, MD, USA). The various formulations used to create the meat analogs are summarized in Table 1. The protein, glycerol, and buffer solution mass ratios in these formulations were held constant at 2:1:7. The polysaccharide concentrations in the formulations were varied from 0 to 4 wt.% by replacing the other three ingredients while maintaining the 2:1:7 ratio.

### 2.2. Mixing Protocol

The buffer solution and glycerol were first blended together using a food processor (10 oz capacity,150 Watts) and then the polysaccharides were added [34,35]. This mixture was then stirred for three minutes to disperse the polysaccharides evenly throughout the liquid mixture. Powdered SPI was then introduced into this mixture and the system was blended for a further two minutes. The samples containing high pectin concentrations (2 and 4 wt.%) required extra time for complete dissolution and so the SPI was added after two hours of mixing for these systems. After mixing, these formulations had textures that varied from viscous pastes to soft agglomerated solids (Figure 1).

### 2.3. Physicochemical Characterization

The thermal and rheological properties of the plant-based formulations were characterized using differential scanning calorimetry and dynamic shear rheology. For the rheology measurements, it was necessary to keep the temperature below the boiling point of water to avoid moisture loss. Thus, parallel-plate rheology tests were performed at 25, 50, and 75 °C and capillary viscometer tests were performed at 75 °C.

#### 2.3.1. Thermal Characterization

The presence of any thermal transitions in the samples was characterized by measuring the heat flow versus temperature profiles of SPI powder and 20 wt% SPI buffer solutions using a differential scanning calorimeter (DSC) instrument (Discovery 250, TA Instruments, New Castle, DE, USA). A small mass (3–5 mg) of sample was placed in an aluminum pan, covered with an aluminum lid, and then sealed. The heat flow was then measured when the temperature was increased from 40 to 200 °C at 5 °C min^−1^ using an empty pan as a reference.

#### 2.3.2. Rheological Characterization

A dynamic shear rheometer (Ares G-2, TA Instruments, New Castle, DE, USA) was used to characterize the shear rheology of the samples using oscillation tests. A measurement cell with an 8 mm serrated parallel-plate geometry was used to avoid slippage. The formulations were placed on the lower plate of the measurement cell to form a 3–4 mm thick sample. The upper plate was then lowered to reduce this thickness of the sample to 0.85–0.9 mm for testing. Excess material oozing out from between the plates was removed using a spatula. A thin layer of paraffin oil was applied to the outer edge of the samples to inhibit the evaporation of moisture from the samples during analysis [36]. Measurements were carried out at three different temperatures (25, 50, and 75 °C). At each temperature, a time sweep test was performed to check the stability of the formulations over a 600 s period using an applied oscillating stress wave with a frequency of 1 Hz and strain of 1%. In addition, an amplitude sweep test was performed to establish the linear viscoelastic regime (LVR) for each formulation and temperature combination at a fixed frequency of 1 Hz. Once the LVR was determined, a frequency sweep test was performed from 0.1 to 30 Hz to determine the frequency dependence of the rheological parameters of the samples, using a constant strain of 1%, which was within the LVR of all the formulations.

The rheological properties of selected formulations were also characterized at 75 °C using a capillary viscometer with a 10 kN load cell (7000 series, Dynisco, Franklin, MA, USA). This device was used because it more closely simulates the conditions samples experience inside a fiber-spinning extruder. A capillary die (L/D 30) with a diameter of 0.762 mm was used for these measurements. The formulations were placed inside the barrel of the instrument and then held for seven minutes to ensure homogeneity and temperature equilibration before starting the measurement sequence. The shear rate was then varied from 30 to 150 s^−1^ using a logarithmic pattern. This range was selected to broadly cover the range of shear rates typically encountered by a sample inside an extruder. It should be noted, however, that higher shear rates might be encountered in the die region of an extruder. In these tests, the forces recorded by the load cell were relatively low and near its error limit.

### 2.4. Extrusion Processing

#### 2.4.1. Extruder Details

A co-rotating twin screw extruder (MC15 HT, Xplore Instruments B.V., Sittard, The Netherlands) to which different circular dies could be fitted was used to produce the fibers. The vertical configuration of the extruder had a circular outlet of 3 mm diameter at the bottom of the barrel. The dies tapered gradually from 3 mm (end of the barrel) to different end diameters depending on design of the die used (0.12, 0.52, or 1 mm). The volume of the barrel with screws in place was 15 mL, as stated by the manufacturer.

The extruder barrel had three heating zones with the die being at the same temperature as the last heating zone. The screws were conical, tapering at the die end with reducing channel depth along the length. Following the screw rotation region was a curved melt flow path of 40 mm length and 3 mm diameter at the end of the barrel, which led the melt to the exit of the die. An image of the extruder with a fitted die is shown in Figure 2.

#### 2.4.2. Extrusion Parameters and Observations

The formulations prepared in the homogenizer were fed to the extruder hopper manually while maintaining the screw speed at 40 rpm. The screw speed was then gradually increased to 150 rpm and the process was allowed to run for 3 min until further observations were recorded. The extrudates could not be collected on the winder accessory due to lack of filament strength. Thus, the extrudates were collected in a Petri dish held 100 mm below the die exit.

The extruder had provision for measuring torque experienced by the screws. At 75 °C, the mass throughput of selected formulations was recorded by collecting the extrudate in a Petri dish for one minute at 150 rpm. The diameter of the extrudates were measured using an optical microscope. Visual observations were made of the surface of the extrudates and of fiber formation during the extrusion process. Two different temperature profiles were used in the extrusion trials (Table 2).

#### 2.4.3. Tensile Characterization

For three formulations (C 2, XG 2, and XG 4) the fibers produced using the 1 mm die at 75 °C were cut into uniform lengths, and then their tensile properties were measured using a tensile meter (KLA T150 Universal Testing Machine, Milpitas, CA) via the grip separation technique. The samples were mounted on a sample retainer card using adhesive tape (see Appendix A). The gauge length of the samples was set at 30 mm. The diameter of each specimen was measured three times and the mean value was calculated and used to determine the applied stress (force per unit surface area). The diameters of all the samples ranged from 975 to 1150 μm. An elongational strain rate of 0.1 s^−1^ was used and the plot of engineering stress versus engineering strain was determined.

## 3. Results

### 3.1. Thermal Characterization

DSC was used to measure the normalized heat flow versus temperature profiles of soy protein in powdered and dissolved form (Figure 3). The literature reports that native soy protein exhibits endothermic peaks around 65 and 90 °C [37]. These peaks are associated with the thermal denaturation of the two major globular protein fractions in soybeans, the 7S fraction (b-conglycinin) and the 11S fraction (glycinin), respectively. These peaks were not observed in the DSC profiles measured in this study, which suggests that the soy proteins had already been denatured. This is probably due to the relatively harsh conditions used to isolate the proteins and converted them into commercial food ingredients, such as solvent extraction, thermal processing, and alkaline solubilization [38,39].

In general, the ability of soy proteins to form aggregate during heating depends on their molecular conformation (native or denatured), with native proteins being more suitable for forming heat-set gels [40].

### 3.2. Impact of Glycerol on the Rheology of Soy Protein Formulations

Preliminary experiments performed using only soy protein mixed with buffer solution showed that the fibers produced after extrusion were extremely brittle and easily fragmented. It is hypothesized that this may have been due to moisture loss during extrusion, as well as due to a lack of plasticization of the protein molecules in the final fibers produced. In fact, this is supported by the apparent weight loss of 3 wt.% in a thermogravimetric analysis (TGA—see Appendix A) of the neat soy protein at 75 °C; it is reasonable then to assume more significant moisture loss is expected with the water-rich SP formulation (70 wt.%—see Table 1). For this reason, the impact of incorporating a food-grade plasticizer and humectant (glycerol) into the formulations on fiber properties was examined. The formulations were mixed using soy protein mixture with systematically increased glycerol concentrations (0, 5, 10, and 20 wt.%) and then their rheological properties were characterized. Figure 4a shows the impact of glycerol concentration on the capillary rheology of SPI formulations held at 75 °C. All the formulations exhibited strong shear thinning behavior, i.e., the viscosity decreased with increasing shear rate. Since capillary rheology probes the pressure-driven extrudability of materials through an orifice die, it is reasonable to predict that these materials will (i) extrude out of the die and (ii) flow more readily at higher shear rates. The viscosity of the formulations increased with increasing glycerol concentration, but all formulations exhibited similar shear thinning behavior.

The impact of glycerol concentration on the frequency dependence of the complex shear viscosity (η*) of the soy protein formulations at 75 °C was also measured (Figure 4b). As with the capillary rheology results, the complex shear viscosity of the formulations increased with increasing glycerol concentration. This effect can be partly attributed to the fact that the viscosity of glycerol is almost three orders of magnitude higher than that of pure water (1 mPa·s at 23 °C) [41]. An increase in viscosity means that a higher force is required to force the sample through the orifice during processing, which would manifest itself as a higher screw torque value. A higher complex viscosity also suggests that a greater pulling force would be required to stretch a fiber after extrusion. The loss tangent (tan δ) was not significantly affected by varying glycerol concentrations. This may have been because glycerol is a Newtonian fluid that may only affect the magnitude of the viscosity, rather than the viscoelasticity of the overall sample.

It should be noted that the presence of glycerol may also impact the strength of the molecular interactions between the protein molecules in the formulations before, during, and after extrusion, which could alter the rheological properties. For example, glycerol has been reported to increase the gel strength of heat-set whey protein gels, which was attributed to its ability to increase the strength of the attractive forces between the globular protein molecules [42]. The addition of glycerol has also been reported to increase the strength of gels formed from myofibrillar proteins for similar reasons [43]. Consequently, the observed increase in viscosity with increasing glycerol concentration observed in the soy protein formulations used in the current study may also have been due to a strengthening of the attractive forces between the protein molecules in the presence of glycerol.

### 3.3. Impact of Polysaccharides on the Rheology of Soy Protein Formulations

The fibers from the soy protein and glycerol mixtures did not exhibit potential for the creation of meat analogs. For instance, they tended to crack and fracture during the extrusion process. For this reason, the impact of adding food-grade polysaccharides (pectin, xanthan gum, and carrageenan) on fiber formation and properties were examined. The rheological properties of the protein–polysaccharide–glycerol mixtures depended on the types and concentrations of polysaccharide added (Figure 5; see Table 1 for the corresponding formulation and codes).

In general, the apparent shear viscosity of the samples decreased with increasing shear rate, which again shows that they exhibited strong shear thinning behavior. At any given shear rate, the apparent shear viscosity of the samples clearly depended on polysaccharide type and concentration. Figure 5d shows the change in complex shear viscosity with increased polysaccharide concentration at a shear rate of 187 s^−1^. The addition of increasing concentrations of pectin did not yield a significant change in viscosity, whereas the addition of increasing concentrations of carrageenan caused a slight increase up to 2 wt%. In contrast, the addition of increasing concentrations of xanthan gum (2 and 4 wt%) and carrageenan (4 wt.%) yielded more pronounced increases in the viscosity. This effect can be attributed to the fact that xanthan gum is much more effective at thickening aqueous solutions than the other two polysaccharides due to its high molecular weight and stiff elongated structure.

For all samples, the storage modulus (G′) was greater than the loss modulus (G″) across the entire frequency, temperature, and composition ranges tested, which implies that the materials exhibited predominantly elastic behavior (see Appendix A). The loss tangent (tan δ) decreased in the following order: pectin > carrageenan > xanthan gum. Overall, these results showed that the three polysaccharides had different effects on the rheological properties of the soy protein composites, which can be attributed to differences in their molecular properties. Pectin has an anionic (carboxyl) linear backbone with neutral side branches attached to certain segments. Carrageenan has an anionic (sulfate) linear backbone. Xanthan gum has a neutral linear backbone with anionic (carboxyl) trisaccharide side groups attached. As a result of these molecular differences, the three polysaccharides would be expected to interact differently with the water, glycerol, and proteins in the composite materials.

For all three polysaccharides, the tan δ value increased with increasing polysaccharide concentration. An increase in tan δ reflects an increase in the viscous character of the material, i.e., the loss modulus (G″) increases relative to the storage modulus (G′). Thus, while all the formulations had predominantly elastic properties (G′ > G″), the introduction of the polysaccharides did increase their viscous character, which may be beneficial for forming fibers with more meat-like properties.

### 3.4. Fiber Formation Using Extrusion

In this series of experiments, the ability to extrude the different protein–polysaccharide–glycerol formulations through a nozzle was assessed. Photographs of different kinds of extrusion behavior that were observed for different formulations are shown in Figure 6.

These experiments were carried out for samples with different formulations (Table 1) and processing temperature profiles (Table 2). The behavior of the formulations during extrusion could be classified into three main categories (Figure 6):i.*Non-extrudable*: these formulations could not be extruded through any of the dies at any temperature.ii.*Poor extrudability*: These formulations could be forced through the die, but the fibers formed were brittle and irregular. These extrudates lacked the ductility and cohesion required for drawing or collection.iii.*Good extrudability:* These formulations could be forced through the die and exhibited enough cohesion to support their own weight up to a certain length and thus could be collected on a plate. Among all the formulations tested, only those formulations containing polysaccharides exhibited this kind of behavior (Table 3).

In the absence of polysaccharide, the soy protein did not form good fibers and would therefore be unsuitable for creating meat analogs. For the large die (1 mm), the samples containing a low amount of pectin could not form good fibers, but all the other samples could. As the die diameter was reduced, it became more difficult to form good fibers. For the intermediate die (0.52 mm), good fibers could only be formed at high xanthan gum and carrageenan concentrations, but for the small die (0.12 mm—not displayed in the table), none of the samples could form good fibers. This phenomenon is due to the greater pressure required to force the samples through a smaller die.

There did not seem to be a strong correlation between the apparent shear viscosity of the samples and their extrudability. For instance, the addition of xanthan gum or carrageenan to the soy protein composites caused a large or small increase in their viscosity, respectively (Figure 5), However, both had similar improvements in extrudability as the concentration was increased (Table 3). Moreover, the addition of increasing amounts of pectin to the soy protein composites sometimes led to a decrease in their apparent shear viscosity but improved their extrudability. This result suggests that some other physicochemical phenomenon must be responsible for the differing impacts of the three polysaccharides on extrudability. A possible reason is the impact of the polysaccharides on the coefficient of friction between the composite materials and extruder elements, as this may lead to ‘single-body-like’ rotation of the material with the screws or to excessive slip. It is possible that the presence of some of the polysaccharides prevented the material from flowing within the post-screw zone of the extruder. In other words, the formulation adhered strongly to the conveying screw rather than flowing out of the die. Based on these observations, piston-type extrusion may work better than a screw-type conveyance, as the capillary rheology experiments discussed earlier suggest that the material can flow out of an orifice die when put under pressure by a piston. Among the three polysaccharides, the fibers formed from the composites containing carrageenan exhibited the best shape retention [44]. Manual handling of the fibers indicated that their firmness increased with increasing polysaccharide concentration.

The temperature profile within the screw extruder also impacted the nature of the fibers formed from the different formulations, with all formulations exhibiting better shape retention at 75 °C than at 50 °C. The carrageenan-based formulations appeared to form the firmest fibers at this higher temperature. Indeed, it was possible to produce continuous extrudates more than 3 m long that could be collected on a plate using the carrageenan-based formulations. The increase in firmness of the samples at higher temperature may be attributed to some unfolding of the globular soy proteins, which increased the attractive hydrophobic interactions between them, leading to the formation of a heat-set gel. Moreover, there may have been some moisture loss, which increased the protein and polysaccharide concentration in the composites, thereby increasing their gel strength. The inability to collect any of these formulations on the winder setup was due to the very low elongational strength and deformability of the extrudates. Clearly, more research is required to optimize the formulation so that fibers can be collected on a winder setup, as this would facilitate the practical utilization.

Figure 7 shows representative stress–strain profiles of those formulations that could be extruded at 75 °C. The mechanical properties and dimensions of the formulations that could be extruded are tabulated in Table 4. The measured ultimate tensile strengths of the XG-based formulations were determined to be in the range of 0.4–0.5 MPa and the strains at break were in the range from 7.5% to 11%. When compared to a pork muscle fiber tensile value [45] (Table 4), the strain-at-break was similar, while the tensile strength and modulus were higher for the XG formulations.

In contrast, the carrageenan-based formulation exhibited more elastic behavior, i.e., a strain at break of 59%, modulus of 1.0 MPa, and strength of 0.2 MPa; the tensile strength was more similar to that of the porcine muscle fiber [45], albeit softer and more elastic. These results highlight the possibility that the tensile properties of an animal muscle can be emulated using soy-protein-based formulations produced by fiber-spinning extrusion, provided that the composition is optimized. In particular, it seems that a long-chain extended polymer, such as a polysaccharide, must be included into the formulation to achieve extrudability and good fiber properties.

It should be noted that the temperatures used in this study were selected (i.e., 75 °C and below) so that there was not significant moisture lost to evaporation during extrusion. In commercial extrusion cooking, the temperatures used are typically much higher than those used in this work, which could impact fiber formation and properties. However, the possibility of creating fibers at lower temperatures may be an advantage in food applications where there are thermally labile substances present, such as omega-3 fatty acids, vitamins, volatile flavors, or carotenoids.

### 3.5. Connecting the Extrusion Behavior to Rheological Properties

In this section, the extrusion behavior of the protein–polysaccharide composites is linked to their rheological properties.

The frequency dependence of the tan δ values of the formulations produced at 75 °C is shown in Figure 8a. Under these conditions, only the two formulations containing low pectin concentrations did not flow out of the extruder. These two formulations exhibited the lowest tan δ values below 0.16. All the formulations that had tan δ values exceeding 0.16 could be extruded from the microcompounder. It is hypothesized that this shift towards a more viscous response of the formulation may be related to the ability of the material to flow out of the extruder; a similar trend was also observed at 50 °C. Such a distinction was not evident in any other rheological parameter (G′ or G″ or η*), implying that the change in the viscoelastic response is a better indicator for SPI extrudability than the absolute magnitude of the viscosity itself.

It has been reported that the storage modulus (G′) of soy protein gels is related to the number of protein molecules involved in the formation of a 3D network structure [46]. In contrast, the loss modulus (G″) is related to the mechanical response of those molecules that do not participate in the formation of a 3D network [40]. In another kind of colloidal system, which consisted of a preceramic polymer system, it was reported that the tan δ value had to be within a specific range (1 to 2.3) to achieve extrudability and that the G′ value should exceed a certain threshold to be drawable [47]. In contrast, the results suggest that for protein–polysaccharide formulations, the tan δ value should be above 0.16.

Figure 8b shows the viscosity plotted against shear rate measured with a capillary rheometer for SP, P 4, XG 4, and C 4. P 4 did not exhibit a notable increase in the viscosity over SP, whereas both XG 4 and C 4 exhibited higher viscosities. While P 4, XG 4, and C 4 were all extruded through the microcompounder, only XG 4 and C 4 yielded long continuous filaments through the smaller orifice die (*d* = 0.52 mm, Table 3). According to Cogswell’s analysis [48], both shear and elongational deformation are attributed to the pressure drop in the entrance of a capillary rheometer. Elongational viscosity is indicative of the melt strength of the extrudate, which is correlated to the extrusion continuity and the post-drawability. Since the shear viscosity under capillary flow is proportional to the extensional viscosity [48], XG4 and C4 likely attained higher extensional viscosities than P4 or SP, which led to more continuous extrusion through the smaller orifice die. This kind of behavior cannot be observed from simple shear deformation experiments using a parallel-plate rheometer.

## 4. Conclusions

This study showed that fibrous materials can be successfully produced using soy proteins in combination with certain kinds of polysaccharides using a scalable fiber-spinning technique, provided the formulation and operating conditions are optimized. Nonetheless, poor extrudability was observed in the case of low polysaccharide (particularly pectin) concentration in the formulation and extrusion through a smaller die. Glycerol as a plasticizer was needed in the formations to inhibit moisture loss during extrusion. Xanthan gum and carrageenan, which are both linear anionic polysaccharides, were more effective at producing extrudable fibers than pectin, which is a branched anionic polysaccharide. This suggests that the molecular and physicochemical properties of the polysaccharides used impacted the formation and properties of the fibers produced by extrusion. The processing behaviors owing to the interplay between the microstructure of polysaccharides were elucidated via rheology; the increase in viscoelasticity (tan δ) as well as capillary viscosity (shear and extensional) were attributed to improved processibility. The results of this study may lead to a new approach for the large-scale production of plant-based meat analogs with a fibrous structure. Ideally, it would be advantageous to produce thinner filaments that could be assembled into bundles of fibers, as these would more closely resemble the structural hierarchy of whole muscle meat products. In future studies, it would be useful to identify an alternative humectant and plasticizer because glycerol may not be label-friendly, as well as having adverse effects on the taste and gastrointestinal fate of meat analogs. Also, further studies are suggested to identify the molecular origin of the different effects of different polysaccharides, especially under uniaxial extensional flow.

## Figures and Tables

**Figure 1 foods-12-03161-f001:**
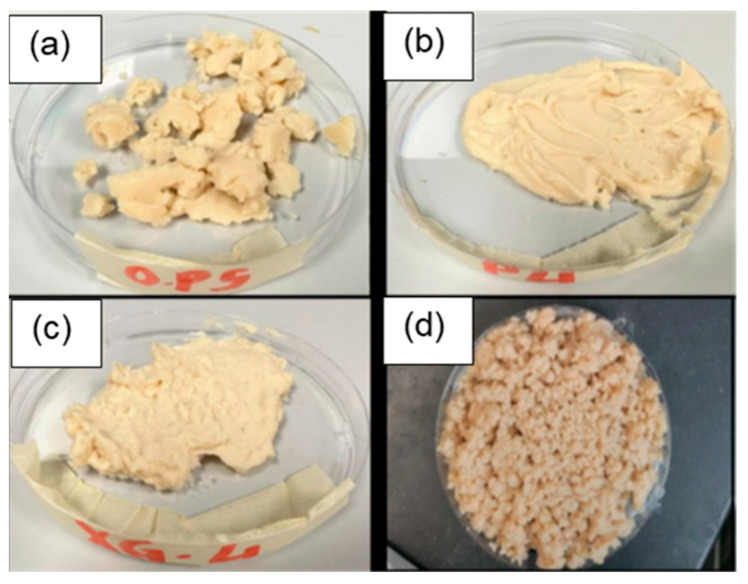
Digital photographs of formulations after being blended in a homogenizer: (**a**) soy protein (SP) only; (**b**) soy protein + 4 wt% pectin (P 4); (**c**) soy protein + 4% xanthan gum (XG 4); (**d**) bottom-right—soy protein + 4% carrageenan (C 4).

**Figure 2 foods-12-03161-f002:**
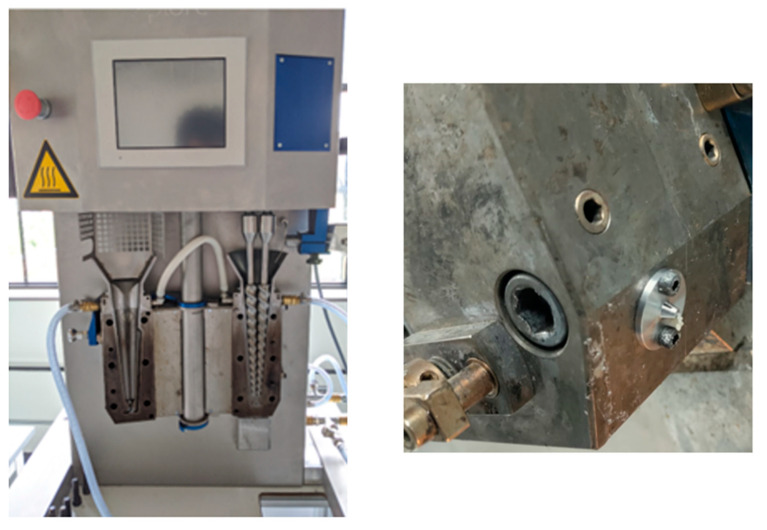
Microcompounder used to produce fibers from the plant-based materials. The **left** photo shows the overall setup, while the **right** one shows an enlarged image of the die used in this study.

**Figure 3 foods-12-03161-f003:**
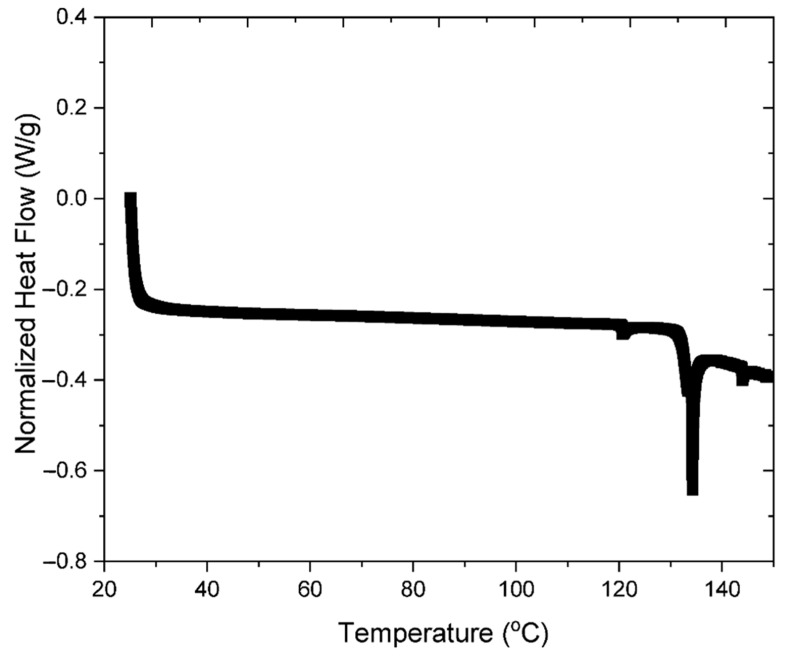
DSC curve showing the normalized heat flow versus temperature profiles of powdered SPI.

**Figure 4 foods-12-03161-f004:**
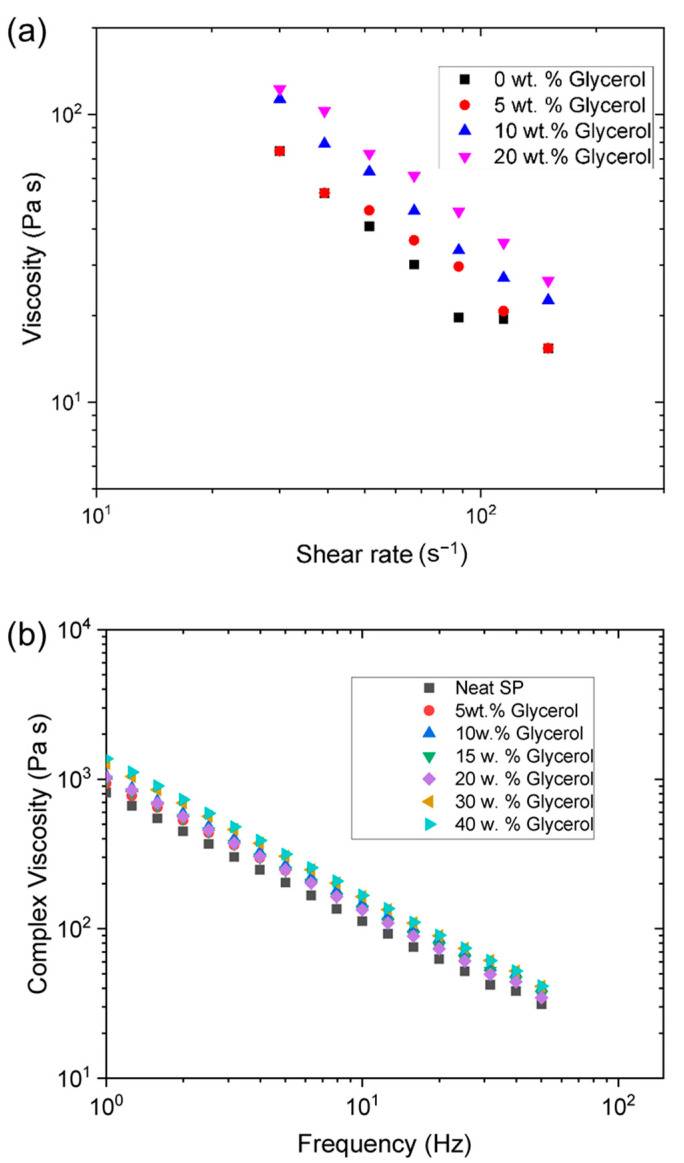
Impact of glycerol on the rheological properties of SPI-based formulations containing no polysaccharide at 75 °C: (**a**) viscosity versus shear rate characterized by capillary rheology; (**b**) complex viscosity versus frequency characterized using dynamic shear rheology.

**Figure 5 foods-12-03161-f005:**
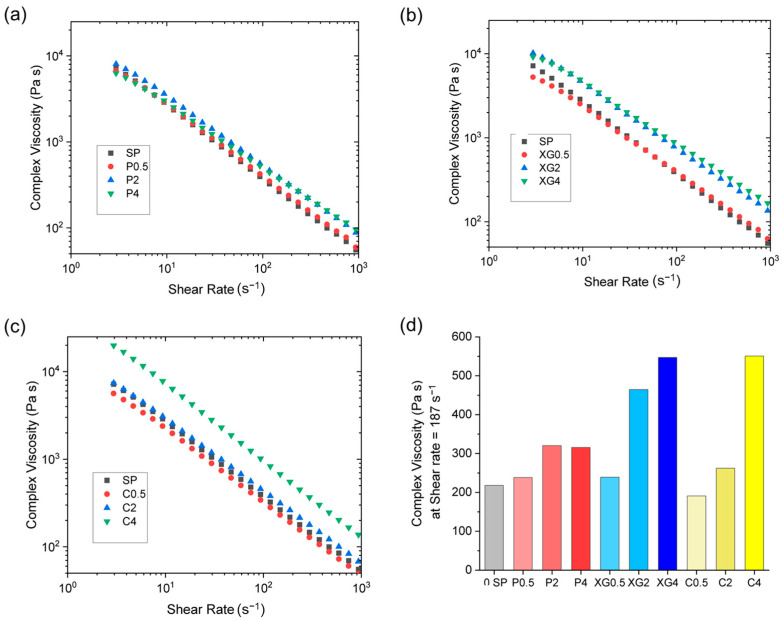
Impact of polysaccharide type and concentration on the apparent shear viscosity versus shear rate for (**a**) pectin, (**b**) xanthan gum, and (**c**) carrageenan composites at 75 °C. (**d**) shows bar chart of the complex viscosity at a shear rate of 187 s^−1^ with varying concentrations of polysaccharides.

**Figure 6 foods-12-03161-f006:**
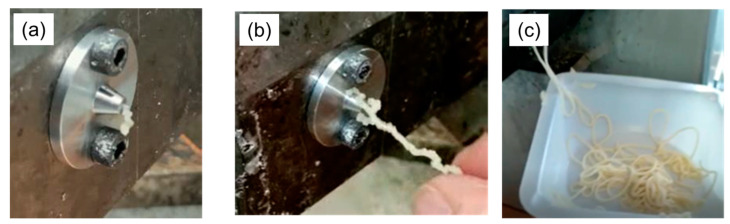
Photographs of different kinds of extrusion behaviors observed in the experiments: (**a**) little or no extrusion; (**b**) formation of brittle irregular filaments; and (**c**) formation of smooth continuous filaments.

**Figure 7 foods-12-03161-f007:**
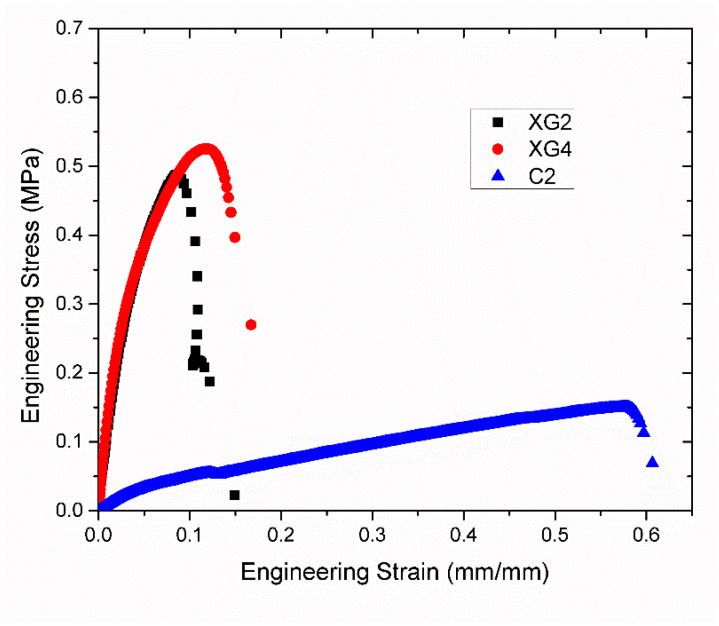
Representative stress–strain curves of protein-polysaccharide fibers measured in a tensile test for formulations XG2, XG4, and C2.

**Figure 8 foods-12-03161-f008:**
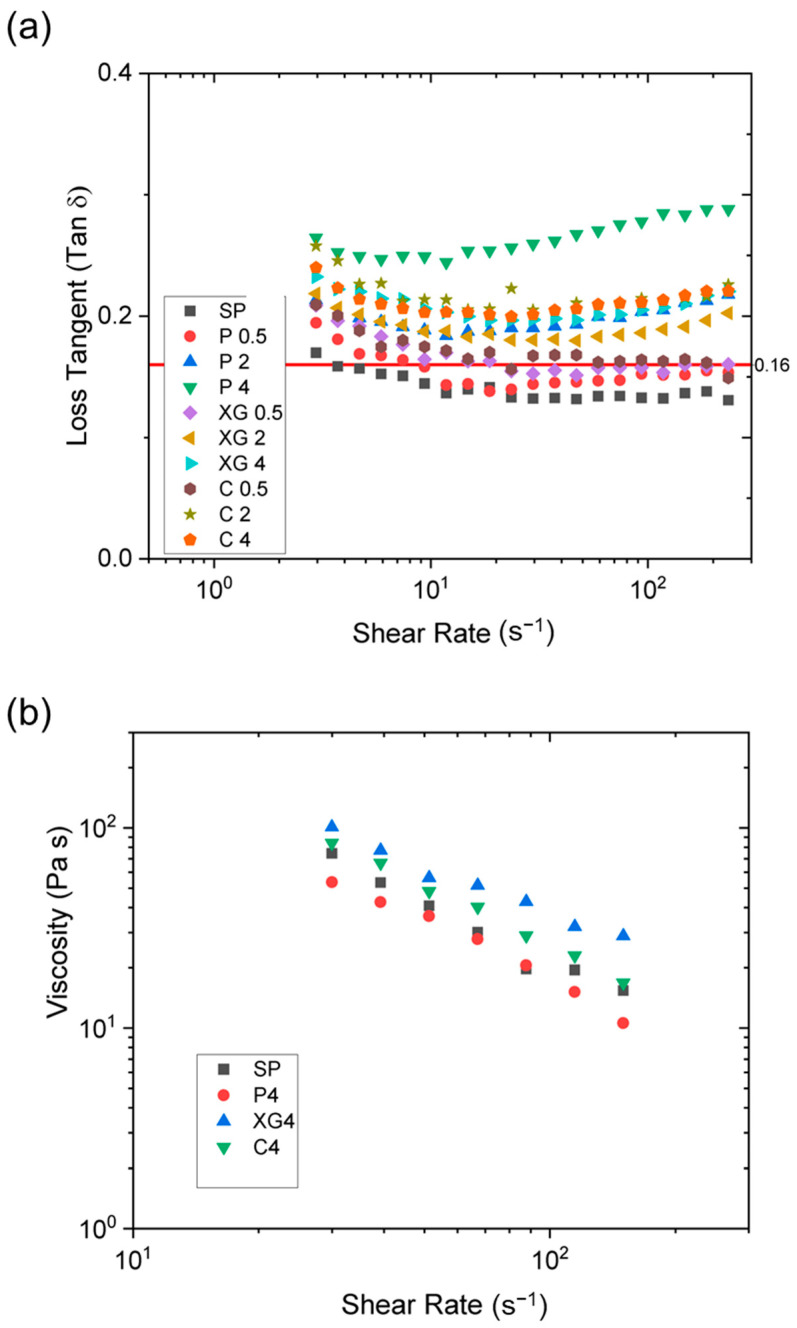
(**a**) Loss tangent as a function of frequency for PS-based formulations using a parallel-plate rheometer. Red horizontal solid line at y = 0.16 indicates the threshold where formulations above were extruded. (**b**) Viscosity as a function of shear rate measured in a capillary rheometer.

**Table 1 foods-12-03161-t001:** Plant-based formulations by weight% of each ingredient used for fiber extrusion. Key: SP = soy protein (no polysaccharides), P = pectin, XG = xanthan gum, and C = carrageenan. The number indicates the wt.% of the corresponding polysaccharide, e.g., P 0.5 = Pectin 0.5 wt.%, XG 2 = Xanthan Gum 2 wt.%, and C 4 = Carrageenan 4 wt.%.

	SP	P 0.5	P 2	P 4	XG 0.5	XG 2	XG 4	C 0.5	C 2	C 4
Soy Protein	20	19.90	19.60	19.20	19.90	19.60	19.20	19.90	19.60	19.20
Buffer (pH 7.0)	70	69.65	68.60	67.20	69.65	68.60	67.20	69.65	68.60	67.20
Glycerol	10	9.95	9.80	9.60	9.95	9.80	9.60	9.95	9.80	9.60
Pectin	-	0.5	2	4	-	-	-	-	-	-
Xanthan Gum	-	-	-	-	0.5	2	4	-	-	-
Carrageenan	-	-	-	-	-	-	-	0.5	2	4

**Table 2 foods-12-03161-t002:** Temperature profiles inside the microcompounder during the extrusion process.

	Zone 1 Temperature (°C)	Zone 2 Temperature (°C)	Zone 3 Temperature (°C)	Die Temperature (°C)
Profile 1	30	40	50	50
Profile 2	55	65	75	75

**Table 3 foods-12-03161-t003:** Summary of extrusion trials for each formulation, temperature profile, and die orifice dimension. Key: a cross (X) means the samples were non-extrudable or exhibited poor extrudability, whereas a tick (✓) means they exhibited good extrudability.

Die—1 mm										
Temperature profile (°C)	SP	P 0.5	P 2	P 4	XG 0.5	XG 2	XG 4	C 0.5	C 2	C 4
30–40–50	X	X	✓	✓	✓	✓	✓	✓	✓	✓
55–65–75	X	X	✓	✓	✓	✓	✓	✓	✓	✓
Die—0.52 mm										
Temperature profile (°C)	SP	P 0.5	P 2	P 4	XG 0.5	XG 2	XG 4	C 0.5	C 2	C 4
30–40–50	X	X	X	X	X	X	X	X	X	X
55–65–75	X	X	X	X	X	✓	✓	X	✓	✓

**Table 4 foods-12-03161-t004:** Tensile data of XG2, XG4, and C2 formulations from the current study compared with reference tensile data of a porcine muscle [38].

Sample ID	Filament Diameter (µm)	Young’s Modulus (MPa)	Ultimate Tensile Strength (MPa)	Strain at Break (%)
XG2	980	11.0	0.433	7.5
XG4	975	15.0	0.496	12
C2	1130	1.0	0.202	59
Porcine Muscle [38]	---	5.04 [38]	0.245 [38]	11 [38]

## Data Availability

The datasets generated for this study are available on request to the corresponding author.

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
