# Peer review of "Optimizing Protein Fiber Spinning to Develop Plant-Based Meat Analogs via Rheological and Physicochemical Analyses"

_foods, 2023, doi:10.3390/foods12173161_

Round 1

Reviewer 1 Report

After reading the manuscript, I realized that the manuscript showed in some parts the scientific rigour wanted, but in other parts I have missed it.

The authors have presented critical evaluation only in some paragraphs.

The references are not exactly current, besides the title and objective could be improved.

Thats why I have written some suggestions below in an attempt to  the paper.

Does it really need fiber and fibrous in the same title ? Think about it. Think about objectives as well. 

- In my opinion, this objective below can be improved further, but it would be better to follow as your title, since  it mentions polysaccharides and soy protein. "Impact of formulation and processing conditions on the creation of individual plant-based fibers assembled from soy proteins and polysaccharides using a fiber-spinning extruder."

I missed the quoting of several numbers of references, you need to check all of them.

L.44-49 -  The paragraph below seems to me that it makes more sense if it comes right after the current L.39."Plant-based meat analogs currently often have advantages over other technologies used to create meat analogs, such as precision fermentation or lab-grown meat [5], [6], [11], [12], because they are less expensive and more scalable. In addition to reducing the negative environmental impacts of the livestock industry, plant-based meat analogs may also have other advantages, including improving human health and animal welfare [9], [10], [13]."

L.59- " Currently" ? - This author is from 2019. 

L.75- What if it was another more fibrous food? Why was soy mentioned here specifically then?

L.78- It was not mentioned about the advantages and previous experiences of other authors, even in other matrices about the successful use of "polysaccharid

L.62-86 - Please avoid "we", prefer in the impersonal. It was assigned, it was examined and so on.

L.58-90- This paragraph needs to be improved, some repetitive sentences. The study rationale needs to be clearer. The objective was mentioned twice and the authors referred the results at this section, which is not usual. This part was like an abstract, it needs to be revised.

L.95-  I missed the explanation or citation of some previous study, why this combination was chosen."High methoxy pectin, xanthan gum, and carrageenan"

L.104 - Protein means "soy protein" ? Pectin means "High methoxy pectin"  Please, more details about it.

L.163- Was extrusion cronologically really 2.3 ?

L.187- It seems to me that the extrusion, physicochemical and rheological analyzes would deserve more highlight in the title and objective of your paper.

L.270 - "The fibers produced using the soy protein and glycerol mixtures did not exhibit appropriate properties for the creation of meat analogs" 

Meat analogs or Meat analogues ?  Are both really correct ? Check your whole paper.

Not succeeding is an important result as well. Should it not be in your conclusion?

Even for hamburgers it could not have worked?

Is it correct to make this statement only with the technological analysis, without a sensory analysis for example ?

L.448- 452- The part on future perspectives should not be inserted in the middle of the conclusion of the paper. After re-evaluating your objectives, revise the conclusion of your study.

Minor editing of English language required

Reviewer 2 Report

Please see the comments attached in the file to revise the manuscript.

The manuscript entitled “Utilization of fiber spinning extrusion to create fibrous plant-based meat analogs: Formulation, processing, and characterization” contains an important study. I have some comments to revise the manuscript.

Thank you very much for providing me the chance to review this manuscript, please see the below comments:

Abstract:

Line 24: derived ingredients. But that the formulation and operating conditions must be optimized to obtain desirable physicochemical and functional attributes of the fibers.

Introduction

Line 75: because these processes produce relatively small………….

Please include some references for recent studies involving fiber spinning extrusion to produce similar structures.

Materials and Methods

It has been explained and written very well.

Results:

Line 208: it has been reported……..

Line 275: on the type and concentration of polysaccharide………

Line 298: these results showed that…….

Line 362: Manual handling of the fibers, what do you mean by it?

Line 371: Please provide a reference.

Line 429: protein-polysaccharide formulations, the tan δ value should………

Conclusions:

Explained very well.

References:

All the references are not uniform, e.g. 7 and 15. The names of journals are in capital letter in one while in other they are abbreviations.

Overall Comments:

  1. All the references should be uniform across the manuscript.
  2. English language must be concise.
  3. The name of all equipment should be mentioned along with their company names and model numbers.

No comments.

Round 2

Reviewer 1 Report

Dear authors, 

After another evaluation of the manuscript, I realized  improvement in the quality of the paper. The authors have accepted some all of my requests. They also have improved  English, which is always useful to ask a native speaker for a final appreciation. They added more authors to better substantiate the methodology and corrected tables and graphs.  This final version of the paper is definitely much better.  Below, still some thoughts:   Avoid using words that are already in the title of your paper as keywords, this won't make it easier for other researchers to find your paper in the future.   Figure 1 - Prefer letters a, b, c, d in figures 

I couldn't find the new references that would have been included, they weren't highlighted in yellow.

"We" still appears on lines 215, 220, 261, 365, 405, 428

Table 1 should also mention "soy protein" and not just protein

L.370 - Why the dot after "Figure 7"?

L.558- Only the author Nascimento et al has a DOI, check and follow the guidelines of  Foods journal.

Minor editing of English language required
